# China’s Trade of Agricultural Products Drives Substantial Greenhouse Gas Emissions

**DOI:** 10.3390/ijerph192315774

**Published:** 2022-11-27

**Authors:** Zheng Meng, Jinling Guo, Kejia Yan, Zhuan Yang, Bozi Li, Bo Zhang, Bin Chen

**Affiliations:** 1School of Management, China University of Mining & Technology (Beijing), Beijing 100083, China; 2School of Management, Xiamen University, Xiamen 361005, China; 3Fudan Tyndall Center, Department of Environmental Science & Engineering, Fudan University, Shanghai 200433, China

**Keywords:** GHG emissions, agricultural products, international trade, time series, China

## Abstract

China’s trade of agricultural products has expanded rapidly over the past two decades, resulting in considerable shifts in greenhouse gas (GHG) emissions worldwide. This study aims to explore the evolution of GHG emissions embodied in China’s trade of agricultural products from 1995 to 2015. The GHG emissions embodied in China’s exports of agricultural products experienced three stages of fluctuation, showing a significant upward trend (1995–2003), a fluctuating trend (2004–2007), and a fall back to the previous level (2008–2015). The embodied GHG emissions in China’s imports were witnessed at times of sustained growth, rising from 10.5 Mt CO_2_-eq in 1995 to 107.7 Mt CO_2_-eq in 2015. The net import of embodied GHG emissions has grown at an average annual rate of 25.1% since 2008. In terms of regional contribution, the distribution of China’s trading partners tended to be diversified. The increasing net imports of oil crops to China resulted in a significant GHG emissions shift from China to the US and Brazil. Asian countries contributed to 76.9% of the total GHG emissions embodied in China’s agricultural exports. The prominent impacts of China’s trade of agricultural products on global GHG emissions provide important implications for climate-related policy choices.

## 1. Introduction

Agriculture fundamentally meets the demands of modern society by supporting various diets, expanding cities, changing living needs, and increasing the world’s population [1]. However, extensive resource consumption and excessive GHG emissions in agriculture have triggered obvious environmental problems and seriously threaten human society [2]. Agricultural activities are the key sources of emissions of anthropogenic greenhouse gas (GHG), and the single largest source of anthropogenic non-carbon dioxide (non-CO_2_) emissions [3]. Agricultural food systems contributed one-third of global anthropogenic GHG emissions in 2015 [3], and non-CO_2_ GHG emissions increased by 32.6% from 4.3 to 5.7 Gt CO_2_-eq yr^−1^ between 1990 and 2015 [2]. Amidst rapid economic growth in developing countries, people’s living standards are gradually improving, and their diets are gradually shifting from grains to meat, eggs, and dairy products [4]. Consequently, total GHG emissions from agriculture may continue to grow. The Paris Agreement aims to strengthen the global response to climate change mitigation this century by keeping the global temperature rise well below 2 °C or even 1.5 °C above preindustrial levels [5]. To achieve the global temperature targets of 2 °C and 1.5 °C, also proposed as China’s 2060 carbon neutrality target, a rapid and far-reaching change in global agriculture is imperative, and agricultural GHG emission reduction will be an important goal.

Agricultural trade plays an important role in global food security and in filling the gap in domestic production. By 2020, global trade in agricultural products had increased over six-fold since 1980 [6]. In international trade the importer gains the agricultural product, while exerting impacts on the environment of the exporter. The GHG emissions related to international trade of agricultural products were 2116.0 Mt CO_2_-eq in 2012, equivalent to 35.1% of the total global direct emissions [7]. The globalization of agricultural trade has led to profound changes in the structure of agricultural trade, most notably in China. With the expansion of the economy, China’s agricultural trade position is gradually shifting from one of net exporters to one of net importers [8]. In 2000, China’s total import of agricultural products was only $19.5 billion, and has increased rapidly to $219.8 billion in 2021, with an average annual growth rate of 12.2% [9]. Meanwhile, China’s agricultural product exports has also witnessed rapid growth, from $16.4 billion in 2000 to $84.4 billion in 2021 [9]. China has become the world’s largest importer and the third largest exporter of agricultural products [8]. CO_2_ emissions from China’s agricultural sector grew from 48.0 Mt to 101.0 Mt over the period of 2000–2019, with an average annual growth rate of 4.2% [10], while China accounted for 26.7% of the global total of GHG emissions in 2019 [11]. In addition, agriculture is the main source of non-CO_2_ GHG emissions, and in 2014 was responsible for 40.2% and 59.5% of CH_4_ and N_2_O emissions in China [12]. The dramatic expansion of China’s trade of agricultural products has led to an increase in its share of global trade, which is accompanied by a huge transfer of GHG emissions between China and other countries.

Along with an increase in the exchange of agricultural products between countries, the related environmental impacts between countries have become obvious. Given the rapid expansion of China’s trade of agricultural products and the continuous development of the global GHG emissions database, accurately quantifying the GHG emissions embodied in the trade of agricultural products and identifying the current role of China in global agricultural emission reduction, is extremely urgent. However, the current production-based accounting methods ignore GHG emissions associated with international trade, and therefore fail to provide a complete picture of GHG emissions for the world and for individual countries. Accordingly, in order to gain a panoramic oversight of global GHG emissions, it is appropriate to take into consideration the transfer of GHG emissions via international trade, especially in trade from countries with strict environmental regulations to countries with lax regulations [13]. The measurement of GHG emissions from agricultural production has received significant attention [14,15,16,17], but the risks of climate change caused by the trade in agricultural products have not yet been fully quantified [18]. Existing studies account for the greenhouse gas emissions of specific agricultural products, such as poultry [19,20], pork [21], vegetables [22,23], and staple foods [24,25]. Friel et al. (2020) [25] confirmed the link between the international trade of agricultural products and GHG emissions, and agricultural trade liberalization led to an increase in total GHG emissions by about 6% in 2015 [26]. Considerable CH_4_ and N_2_O emissions in international agricultural trade limit the development prospects of the livestock industry under climate changes [27]. The embodied GHG emissions in the international trade in livestock products cause developing nations to face increasing difficulties in fulfilling emission reduction obligations, which also impedes local economic development [28]. Lin et al. [29] used the EIO/LCA method to account and analyze China’s food production from 1979 to 2009 and determine the carbon emission factors (CEFs) for 15 kinds of food products in China. Accurate accounting of the greenhouse gas emissions in the international trade in livestock products provides a basis for drawing a red line under climate change [30]. Moreover, there exist numerous studies on the GHG emissions produced by China’s trade of agricultural products with foreign countries, such as Germany [31], the U.S. [32,33], Brazil [34], and India [35]. Previous studies provide valuable empirical evidence for managing the transfer of GHG emissions from agricultural trade [36]. The GHG emissions embodied in agricultural products which have resulted from changes in China’s agricultural trade trends remain largely unknown. Thus, in order to maximize China’s climate change benefits from agricultural trade, we must map the trends in China’s GHG implicit in agricultural trade and identify the factors driving structural changes.

This study aimed to capture the trade-related GHG emissions of China’s agricultural products and to evaluate their structures. The corresponding composition and contribution ratio of numerous foreign countries were identified, in order to reveal their impacts on the GHG emissions of China’s trade of agricultural products. The results can help policymakers quantify the amount of GHG emissions of different kinds of agricultural products and determine appropriate emission reduction targets. In addition, a database is provided to assist stakeholders in optimizing the analysis of the structure of trade, domestic production, and domestic consumption. This database will facilitate international collaboration in efficient GHG emission reduction in China’s agricultural sector.

The rest of the paper is organized as follows: Section 2 introduces the methods for GHG emissions estimation and data sources. The results of GHG emissions, embodied emissions in exports and imports in inter-regional trade are presented in Section 3. Section 4 discusses the GHG emission characteristics of China’s agricultural trade and compares different GHG emission factors. Concluding remarks are addressed in Section 5. The research framework of this study is illustrated in Figure 1.

## 2. Data and Methodology

### 2.1. Accounting Method

Accounting methods in international trade in agricultural products can separate production from consumption. Production-based accounting generally allocates GHG emissions to the production area [37], and therefore fails to trace the inter-country transfer of GHG emissions, which leads to the dilemma of “inequity in emission rights”. This study aimed to provide a comprehensive account of the trade of agricultural products related to the embodied GHG emissions in China and identify the factors driving structural changes with the help of carbon emissions factors. Referring to previous studies [29], the agricultural GHG emission boundaries in this study were carbon dioxide, methane and nitrous oxide. The categories of GHG emissions in agricultural production activities (Appendix A) are based on FAO’s GHG calculation guidelines and are shown with the GHG they produce. Land-use and feed-processing activities are not included in agricultural production activities, referring to previous works [7,38]. CEF considers the GHG emissions occurring during the whole production chain of commodities and services, and therefore enables consumption-based accounting to account for the GHG emissions embodied in the commodities and services. The agricultural GHG for countries (regions) are categorized by agricultural activities using the Food and Agricultural Organization of the United Nations (FAO) [8] guidelines, rather than specifically by product [38]. It should be noted that the wide disparity in agricultural technology and resource endowments among countries (regions) brings about a misalignment with the FAO guidelines regarding CEFs and therefore misguides policy design. The study was meant to cover all countries with agricultural trade with China, and the data uncertainty caused by FAO’s CEFs may distort the final results. Most crucially, the CEFs provided by the FAO is insufficient for investigating trends over time and needs to be properly calibrated. By taking uniform accounting into consideration, Hawkins et al. (2016) [38] improved a set of CEFs to evaluate GHG emissions embodied in China’s trade of agricultural products. It enabled accounting for direct and indirect GHG sources over time and helped determine how China’s changing preferences for crops and livestock products affected GHG emissions. As a result, CEFs in Hawkins et al.’s study [38] were chosen as a benchmark due to its comprehensive scope of China’s trade of agricultural products, and long time-series data. In order to strengthen the credibility of the results, this study attempted to determine the GHG emissions embodied in agricultural products with two sets of CEFs from the FAOSTAT [8] and Hawkins et al. [38], respectively, and compared the results. Subsequently, the time frame was extended to provide a set of embodied GHG emissions for China’s agricultural products.

This study employed the Intergovernmental Panel on Climate Change (IPCC)’s Tier 1 methodology [37] to determine the embodied GHG emissions of China’s trade of agricultural products in a consistent and comparable manner. For China, GHG emissions embodied in imports and exports were determined by multiplying the trade volume of commodities and services with the corresponding CEFs of commodities and services in the exporting country:(1)$$EEI=\sum_{s}\sum_{j}^{}(CEF_{s,j}^{}\times R_{s,j}^{import})$$
(2)$$EEE=\sum_{s}\sum_{j=1}^{}(CEF_{j}^{}\times R_{s,j}^{export})$$
where *EEE* means the GHG emissions embodied in China’s agricultural product exports; *EEI* represents the GHG emissions embodied in China’s agricultural product imports; *CEF_s,j_* and *CEF_j_* denote the GHG emissions factors (unit: t of CO_2_-eq/t product) of agricultural product *j* in a foreign country or region *S* and China, respectively; $R_{s,j}^{import}$ represents China’s import volume of product *j* (unit: t) from a foreign area *S*; and $R_{s,j}^{export}$ stands for China’s export volume of product *j* (unit: t) to a foreign area *S*.

Furthermore, *EET* denotes the GHG emissions embodied in China’s total trade of agricultural products. *EEB* represents the GHG emissions embodied in China’s net trade of agricultural products.
*EET* = *EEI* + *EEE*
(3)
*EEB* = *EEI* − *EEE*(4)

### 2.2. Data Sources

The data pertaining to China’s trade of agricultural products from 1995 to 2015 were derived from the China Customs Statistical Yearbook [39], which contains commercial activities between mainland China and 219 other countries (regions). In total, 329 products were counted, covering 15 categories, including rice, wheat, corn, fruit, oil crops, sugar, beans, tubers, vegetables, milk, beef, mutton, pork, poultry, and eggs, as depicted in Appendix A. Miscellaneous products that were difficult to categorize, such as baby food and indistinguishable food, were excluded from the study. Several issues relating to the data should be noted. (1) Unit problem: the unit of measurement for live animals is counted by the numbers of animals, while most other products in the trade of agricultural products are counted by tons [39]. For unity, this study converts the measurement unit of live animals into tons according to the average weight [38]. (2) Problems caused by history: although the data from Yugoslavia, the Soviet Union, and South Sudan can be found in the database, data for these countries are ignored in this study.

## 3. Results

### 3.1. GHG Emissions Embodied in China’s Agricultural Product Exports

The GHG emissions embodied in China’s export of agricultural products are summarized in Figure 2a–c. There were considerable GHG emissions associated with food crops, but livestock products-related were relatively flat during the accounting period. During 1995–2015, the average *EEE* of China’s agricultural products was 16.6 Mt CO_2_-eq, and the *EEE* in 2015 was 11.4 Mt CO_2_-eq, less than half of 28.3 Mt CO_2_-eq in 2008. The *EEE* of food crops and livestock were 9.8 Mt CO_2_-eq and 1.6 Mt CO_2_-eq in 2015, respectively, and the dominant *EEE* of agricultural products gradually shifted from corn and rice to vegetables and fruit, as illustrated in Appendix A.

Figure 3 presents the composition of food crops and livestock products among China’s agricultural exports. On average, food crops contributed 86.5% of agricultural *EEE*. From 1995 to 2003, the proportion of *EEE* from food crops increased steadily and peaked at 90.9% in 2003. Subsequently, the proportion declined and since 2005 has fluctuated at around 85.0%. The internal composition has changed tremendously, with fruit and vegetables gradually replacing grains as the core elements. Inconsistent with the general trend of *EEE*, livestock products peaked in 2006 and then declined to 1.6 Mt CO_2_-eq (see Appendix A). Pork, poultry, and eggs played dominant roles in the *EEE* of livestock products, taking up 44.0%, 23.4% and 20.2% in 2015, respectively.

Figure 4 depicts the major *EEE* flows associated with China’s agricultural products in 2015, which were mainly distributed throughout Asia, Europe, and North America. Especially countries (regions) such as Mongolia, Hong Kong, Singapore, South Korea, Japan, Russia, Malaysia, Indonesia, and the U.S. acted as China’s main *EEE* partners. Mongolia contributed 47.1% (2.5 Mt CO_2_-eq) of the *EEE* dominated by fruit exports, followed by Singapore with 20.3% (1.1 Mt CO_2_-eq). Detailed information about China’s *EEE* of agricultural products by countries (regions) can be found in Appendix A.

### 3.2. GHG Emissions Embodied in China’s Agricultural Product Imports

Figure 5a illustrates the *EEI* of China’s agricultural products during 1995–2015. On average, the *EEI* of China’s agricultural products reached 32.4 Mt CO_2_-eq. The *EEI* of China’s agricultural products displayed a “V-shape” from 1995 to 2000, which reached 10.5 Mt CO_2_-eq in 1995 and 22.0 Mt CO_2_-eq in 2000. As it was impacted by the international financial crisis in 2008, a significant drop in *EEI* was witnessed. Actually, the *EEI* of food crops and livestock products dropped to 18.8 Mt CO_2_-eq and 7.8 Mt CO_2_-eq in 2009, see Figure 5b,c. Appendix A indicates that *EEI* grew rapidly in 2010 and reached its historical peak at 107.7 Mt CO_2_-eq in 2015, with an average annual growth rate of 16.2%. The major contribution to China’s *EEI* of agricultural products in 2015 came from oil crops (40.1%), rice (24.9%), pork (8.6%), wheat (7.7%), mutton (5.5%), fruit (2.5%) and milk (2.3%).

The specific structural changes are presented in Figure 6. Food crops and livestock products grew continuously, and detailed information about this growth is shown in Appendix A. Pork, beef, and mutton played dominant roles in the *EEI* of livestock products, accounting for 36.3%, 29.0% and 23.2% in 2015, respectively.

Figure 7 demonstrates the top ten *EEI* flows related to agricultural products for China in 2015. China’s major *EEI* partners were North America, South America, Oceania and Asia, together accounting for 91.0% of the total *EEI*. A significant disparity exists between agricultural products when it comes to the concentration of *EEI* (the ratio of total *EEI*/*EEI* of China’s top 10 trading partners). The concentrations of *EEI* for each agricultural product are as follows: mutton 99.8%, beef 99.6%, tubers 99.0%, oil crops 96.3%, rice 94.7%, vegetables 92.2%, poultry 89.9%, wheat 86.0%, beans 84.5%, milk 82.0%, fruit 57.1%, pork 47.4%, sugar 37.9%, eggs 24.3%, and corn 11.2%. As China’s largest trading partner, the U.S. accounts for 28.9% of China’s agricultural product *EEI*, close to the combined total of its second (Brazil) and third (Australia) trading partners. Brazil accounted for 46.3% (20.4 Mt CO_2_-eq) of China’s *EEI* in oil crops, followed by the U.S. (27.6%, 11. 9 Mt CO_2_-eq) and Argentina (11.2%, 4.8 Mt CO_2_-eq). The U.S. took a stable position as the most significant import source of agricultural products from China. Detailed data is shown in Appendix A.

### 3.3. GHG Emissions Embodied in the Net Trade of Agricultural Products

Figure 8a presents the trend in China’s net import volume from 1995 to 2015, changing within a broadband between −3.8 Mt (in 1997) and 126.8 Mt (in 2015). A turbulent trend was witnessed during 1995–2003, with China’s role in switching from a net importer to a net exporter (see Appendix A). Figure 8b depicts the *EEB* in China’s agricultural products, which increased fiftyfold during the period, making an increase from 1.9 Mt CO_2_-eq in 1995 to 96.3 Mt CO_2_-eq in 2015. Overall, the trend of *EEB* was consistent with the net import volume, whereas *EET* noticeably fluctuated with the trade volume, as indicated in Appendix A. *EEB* displayed an accelerated net import trend. It could be rationalized that the rise in the renminbi (RMB) increased the costs of agricultural products and further weakened the price advantage of China’s exports. The *EEB* of corn, wheat and rice, as it changed from negative to positive, coincided with China’s transformation from an agricultural to an industrialized country. Fruit exports, as an essential component of *EEB*, also declined slightly over the past 21 years. China has the world’s largest urban population, and around 70.0% of its people permanently live in cities where high-quality animal protein is in high demand [8]. The *EEB* of livestock products has soared in recent years and reached 23.8 Mt CO_2_-eq in 2015. Thus, as expected, from 1995 to 2015, the rapid socio-economic development of China’s population has led to dramatic changes in EEB.

Figure 8c lists the structure of *EEB* of major countries (regions) during the period concerned. The U.S. was China’s largest *EEB* source, reaching 181.3 Mt CO_2_-eq. Negative EEB values for Japan, Korea, Mongolia, Hong Kong and Singapore represent China’s export of embodied GHG emissions to these countries (regions). Trading partners tended to be diversified during the period 1995–2015, indicating that the structure and policy of China’s trade of agricultural products were desirable (see Appendix A). China mainly imported oil crops from the United States, Brazil and Argentina, resulting in positive *EEB* between China and the United States, Brazil and Argentina, accounting for 58.5%, 97.4% and 95.8%, respectively. It is worth noting that uncontrolled importation of oil crops due to the price advantage may have had a negative impact on China’s domestic crop structure. Japan is a long-standing trading partner with China, and the net embodied GHG emissions from China to Japan were 25.8% for rice, 19.8% for corn, 19.8% for oil crops, and 15.1% for vegetables. Figure 8d lists separately the *EEB* distribution of China’s major trading partners in 2015. Hong Kong was China’s largest net embodied GHG emissions export region, mainly consisting of 37.6% pork, 19.2% vegetables, and 13.3% fruit. The negative values of *EEB* from China to Mongolia and Singapore were chiefly reflected in fruit exports, reaching 88.5% and 70.9%, respectively. Rice and oil crops accounted for a significant proportion of *EEB* from the U.S. to China, at 53.8% and 38.3%, respectively (see Appendix A). The diversified trade in agricultural products could help reduce China’s dependence on a specific country. Given this, China’s endeavors in maintaining diversification and mutually beneficial development should be continued.

## 4. Discussion

### 4.1. GHG Emissions Associated with China’s Trade of Agricultural Products

With the growing demand in China for agricultural products, the trade in agricultural products will continue in the future, which will result in increasing embodied GHG emissions in trade. We accounted for the trend in China’s agricultural trade with other countries based on the classification of agricultural products in Appendix A.

The total export volume of China’s agricultural products increased from 1995 (5.5 Mt) to its peak in 2003 (30.5 Mt). The subsequent fluctuations flattened out to 13.6 Mt after 2008. The export volume of rice increased from 0.3 Mt to 3.9 Mt over 1995–1998 and continued to slightly fluctuate in the following years. The export volume of grain gradually declined as described in Appendix A. The fluctuation in grain exports has affected the variations in the export structure. The reduction in the corn export quota and the cancellation of export subsidies resulted in a huge reversal, from 16.5 Mt in 2003 to 2.4 Mt in 2004. In contrast, vegetables have consistently taken up a significantly dominant position in recent years. Vegetables reached 6.2 Mt in 2015, with an average annual growth rate of 7.6%. China’s vegetable planting areas covered 22.0 million hectares in 2015, with a market share of nearly 11.0% of the overall international trade, revealing China’s strength in vegetable export operations [9]. Overall, the export volume of livestock products continued to slightly fluctuate (from 0.9 Mt to 0.7 Mt) in the period concerned. Pork and poultry achieved the average annual trade volumes of 0.3 Mt. The fluctuation ranges of beef, mutton, milk, and egg were insignificant and the average annual export trade volume was 0.3 Mt, 0.1 Mt, 0.5 Mt, and 0.8 Mt, respectively. Appendix A reveals that the proportion of China’s agricultural product exported to Asia was close to 70%, followed by exports to North America and Europe. The countries (regions) that frequently traded with mainland China in 2015 were the U.S. (3.0%), Thailand (6.4%), Indonesia (6.4%), Russia (6.6%), Malaysia (7.5%), South Korea (7.9%), Japan (9.4%), Vietnam (10.2%) and Hong Kong (13.2%), with the overall export volume amounting to 70.4%.

After China joined the World Trade Organization (WTO), the importation of agricultural products reached a high-growth rate (average annual growth rate of 15.4%). During the period concerned, China’s imports of food crops continued to soar, while the imports of livestock products rose slowly, as presented in Appendix A. The import volume of food crops noticeably fluctuated. Due to the decrease in international wheat prices, the import volume of wheat rose between 2008 and 2015. Over the past 21 years, oil crop imports have risen by 209 times with an average annual growth rate of 29.0%, as portrayed in Appendix A. In general, the import volume of livestock products has increased since 2000. The demand for meat and dairy products has greatly increased in China; however, inadequate domestic production has failed to meet this demand. As a result, the importation of milk increased continuously from 0.1 Mt in 1995 to 1.6 Mt in 2015, with an average annual growth rate of 17.0%. Before 2000, Asia, North America, and the European Union were as the main importers of China’s agricultural products, taking up 46.6%, 23.6% and 20.8% of China’s total imports, respectively. After the Millennium, the distribution of the main import sources of agricultural products gradually shifted from the developed countries (e.g., the U.S., Japan, and the EU) to a diversified pattern. Thus, the market share concentration has gradually been progressive. The specific data of major import trading partners is delineated in Appendix A. The main import trading partners of China’s agricultural products in 2015, comprising Brazil, the U.S., Argentina, Thailand, Australia, and Canada, generally take up 82.5% of its imports.

In theory, importing low-CEFs agricultural products should generate climate benefits [40], as increased imports of agricultural products replace domestic production and reduce GHG emissions from domestic agricultural production in China. China’s growing trade deficit in agricultural products could help hinder the continuous growth of GHG emissions caused by industrialization [41]. Thus, EEB also indicates emission reductions from China’s agricultural trade. This study confirmed that the reduction effect of embodied GHG emissions in China’s trade of agricultural products was continuously strong (approximately 96.3 Mt CO_2_-eq in 2015). Notably, GHG emission reduction potential in trade was only a small part of the agricultural emissions (total 830.0 Mt CO_2_-eq in 2014) [34]. GHG emissions embodied in China’s agricultural imports per capita were only 70 kg CO_2_-eq in 2015, less than half the level of the U.S. [42], even though China has become the world’s largest importer of agricultural products. Compared with the meat-based food consumption structure, China’s grain-based diet structure is more sustainable [27]. Domestically produced staple foods can ensure that the country’s population will continue to eat sustainably (see Table 1); however, a trend in the net import of oil crops is increasingly apparent. On the premise of ensuring the security of the food supply, the international trade policy of agricultural products and domestic environmental policies should be coordinated, and the importation of low-emission-intensity agricultural products should be promoted, which is conducive to a reduction in GHG emissions from domestic agricultural production [43].

### 4.2. CEFs of Agricultural Products

Differences in geographical location, climate conditions, and resource endowments have caused a gap in the CEFs of agricultural products among different countries (regions), which finally affect the embodied GHG emissions of international agricultural products. Due to differences in production technology and emission standards, low GHG agricultural production has been promoted in developed countries. Moreover, developed countries tend to import agricultural products with high CEFs, to create favorable conditions for domestic carbon neutralization through international trade.

The CEFs of China’s agricultural production are generally lower than that of other developing countries (See Appendix A). After the “Eighth Five-Year Plan” of China’s national economic development first proposed “ecological agriculture”, the State Council issued the “National Ecological Environment Construction Plan” in 1999. This was followed by a series of sustainable development policies in agriculture, which have achieved significant effectiveness [44,45]. The export volume in 2015 was 2.3 times that in 1995, while the embodied GHG emissions increased merely by 32.4%. The main reason behind this is the significant decrease in the CEFs of China’s agricultural products, with the average CEFs of exported agricultural products having dropped from 1.6 t of CO_2_-eq/t per product to 0.9 t of CO_2_-eq/t per product. China has set a positive example for the world by proactively promoting low-GHG agriculture and improving agricultural technology to reduce GHG emissions on the production side [46]. Relatively low prices have made China’s agricultural products extremely competitive in the market [9], which is conducive to global GHG reduction. Furthermore, the gradual rationalization of the structure of China’s trade of agricultural products has boosted the reduction in GHG emissions in international trade [47].

Mongolia and India both exported 0.1 Mt of agricultural products to China in 2015, and the two countries generated significantly different amounts of GHG emissions. India emitted 0.1 Mt CO_2_-eq of GHG, merely 1/50 of that of Mongolia. From the perspective of the production side, the preferential importation of agricultural products from India can reduce global GHG emissions while satisfying domestic demand. The average CEF of rice in different countries was the highest among food crops, reaching 3.6 t of CO_2_-eq/t product. China’s main sources of imported rice have gradually become concentrated in Australia, Vietnam, and Thailand, all of which have a CEF below the average level. This demonstrates that the transformation of China’s import structure has facilitated global total emission reduction and a reduction in the production side of emission reduction. The CEF of fruit in inland and arid countries was found to be higher than that of coastal and water-rich countries. For example, the CEF of Mongolia’s fruit production was 15,917.0 times that of the Maldives.

### 4.3. Comparison of Different CEF Benchmarks

The changes in the CEFs of three staple grains (corn, rice, wheat) and livestock products under different benchmarks were used to reveal the effect of CEF fluctuations on the *EEE* and *EEI* of agricultural products. The CEFs from the FAOSTAT [8] and the CEFs corrected by Hawkins et al. (2016) [38] were employed in this study and the results were compared (see Appendix A), as presented in Figure 9a. According to the CEFs provided by FAO, the *EEE* of China’s staple food and livestock products from 1995 to 2015 were on average 6.1 Mt CO_2_-eq and 0.9 Mt CO_2_-eq, respectively. For the three staple grains, the CEFs of the two groups were similar, notably the CEFs of wheat, leading to extremely similar results for both. Significant gaps existed in the *EEE* of livestock products, as the wide differences in the CEFs of eggs and beef led to this result. In Figure 9b, the trend of *EEI* based on the CEFs of FAO showed obvious differences from the corrected CEFs by Hawkins et al. [38]. The CEFs of rice in Vietnam, Thailand, and Pakistan provided by the FAOSTAT [8] were higher than that revealed by Hawkins et al. [38] in 2009, thereby leading to different *EEI* results. *EEI* trade in livestock products was relatively stable. Though the results based on the two sets of CEFs showed discrepancies to a certain extent, the overall trend was convergent. The small difference in CEFs for China’s major importing countries of livestock products (e.g., the U.S., Brazil, Argentina, and New Zealand) is the main reason for this result. For example, in the two groups, the CEFs of beef and mutton in Argentina were equivalent.

### 4.4. Limitations

As discussed above, any alterations in key CEFs would diffuse rapidly over the results (as shown in Figure 9). Accurate CEFs of agricultural products can improve the accuracy of results, therefore helping policymakers to make feasible decisions. Notably, in specific situations, the confirmation of accurate CEFs for agricultural products could avoid leading policymakers to have impractical expectations of higher errors in data results, therefore addressing this issue is a feasible way of optimizing CEFs benchmarks. In particular, the lack of data from African countries caused uncertainty in our results, hindering specific analysis that could be critical for both domestic and international mitigation planning.

In this study, re-importation of commodities and services was not considered. China mainly imports oil crops and exports vegetables and fruit. The correlation between import and export of agricultural products is weak and the effect of re-importation is insignificant. The re-importation of agricultural products exerts a relatively small impact on China’s total GHG emissions. However, re-importation accurately impacts the GHG emissions of exporters. Nevertheless, the embodied GHG emissions of raw materials should be considered by the raw-material-exporting countries. Therefore, the embodied GHG emissions in imported agricultural products should be theoretically analyzed. On that basis, the LCA nesting theory could be combined with this research to reveal the embodied GHG emissions. Since data of the embodied GHG emissions in the global trade of agricultural products is rare, the re-importation effect should be analyzed in-depth in future research.

## 5. Conclusions

This study was conducted to depict the GHG emissions embodied in China’s trade of agricultural products. The time-series results comprehensively revealed the spatiotemporal evolution characteristics and flow patterns of GHG emissions associated with China’s trade of agricultural products. The exported embodied GHG emissions in 2015 were 11.5 Mt CO_2_-eq, of which the main contributors were fruit (46.2%), vegetables (22.2%), rice (7.5%), oil crops (5.7%), tubers (1.8%), and beans (1.6%). From 1995 to 2015, the GHG emissions related to China’s agricultural product imports were 32.4 Mt CO_2_-eq per year, dominated by oil crops. The U.S. contributed the most to China’s imported emissions, accounting for nearly 28.0% and increasing slowly. China’s increased trade deficit in oil crops is noteworthy. It is time for China to adjust its trading structure of agricultural products. Moreover, a shift in the Chinese population’s diet to more vegetables and fruit, and less meat (e.g., pork, beef, and mutton) and beans should be encouraged, and this can reduce the demand for imported agricultural products.

The necessity for accurate emission estimates and the link between global agricultural GHG emissions and international trade needs to be highlighted by policymakers and the climate change community [48]. For China, an improvement in domestic production and an adjustment in trade policies are required to realize the mitigation of GHG emissions associated with China’s trade of agricultural products. It is imperative to draw on the experience of developed countries in the international import of agriculture products (such as the E.U.’s carbon border adjustment mechanism) and to develop effective solutions for agreements involving China’s export of agriculture products (such as free trade agreements), as well as to reduce the embodied GHG transfers in China. The results of this study provide feasible ideas for consistent analysis of the embodied GHG emissions in the trade of agricultural products between different countries. More data sources and models are critical to help policymakers regulate trade structures and achieve mitigation goals (even to achieve Pareto optimality).

## Figures and Tables

**Figure 1 ijerph-19-15774-f001:**
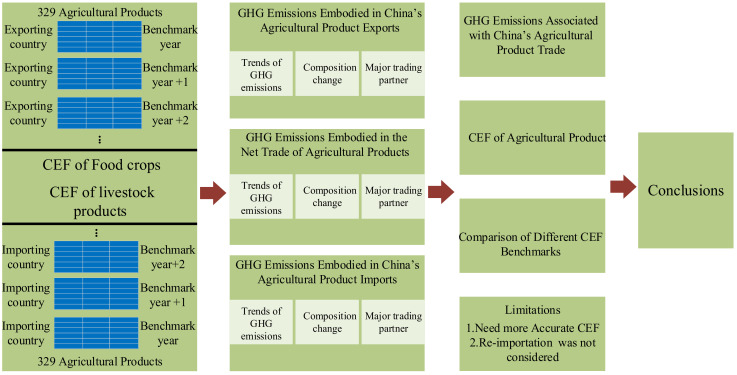
Research framework of this study.

**Figure 2 ijerph-19-15774-f002:**
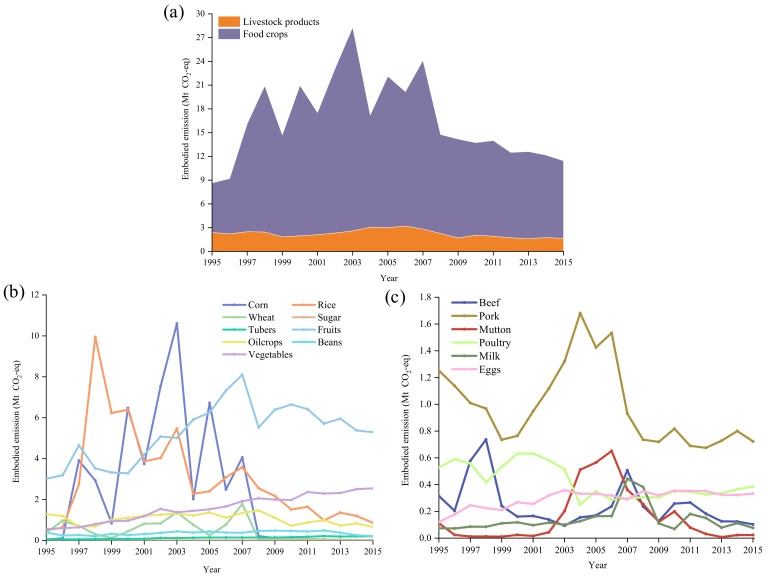
GHG emissions embodied in China’s export of agricultural products: (**a**) total agricultural products during 1995–2015; (**b**) food crops during 1995–2015; (**c**) livestock products during 1995–2015.

**Figure 3 ijerph-19-15774-f003:**
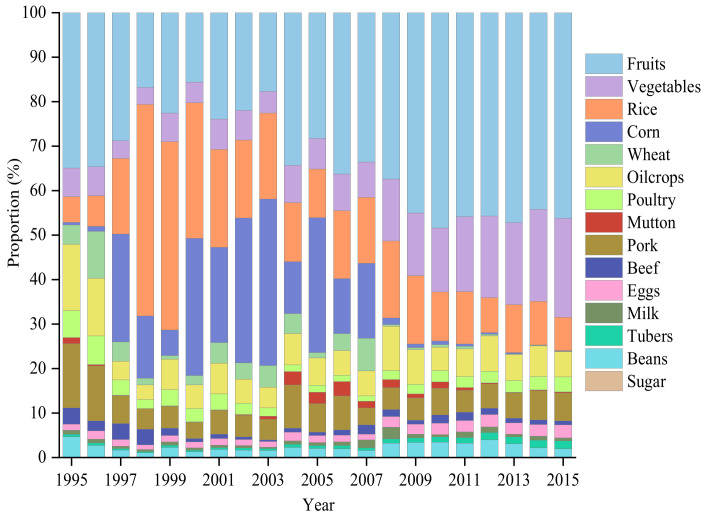
Compositions of GHG emissions embodied in exports by the type of agricultural products during 1995–2015.

**Figure 4 ijerph-19-15774-f004:**
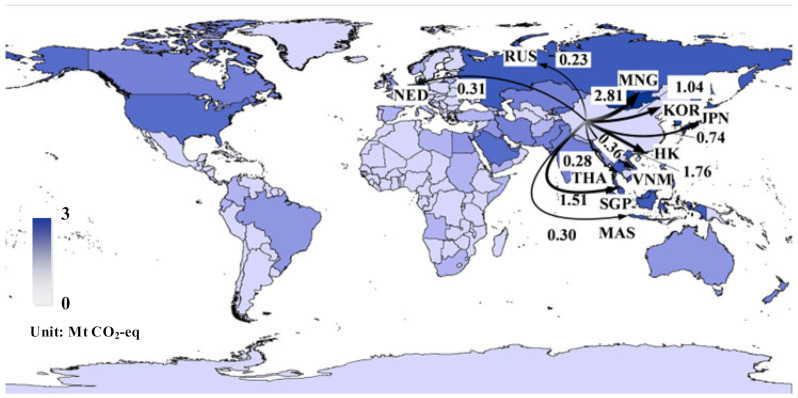
Major GHG emission flows embodied in China’s export of agricultural products in 2015.

**Figure 5 ijerph-19-15774-f005:**
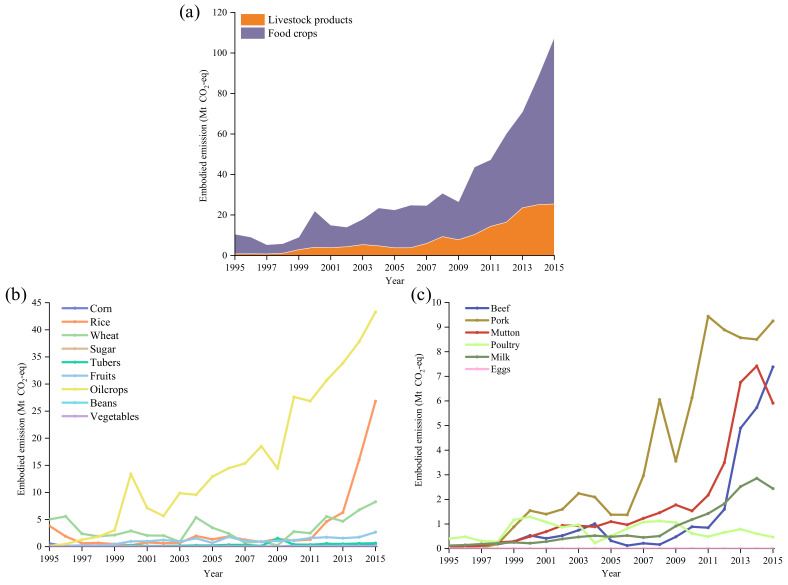
GHG emissions embodied in China’s import of agricultural products: (**a**) total agricultural products during 1995–2015; (**b**) food crops during 1995–2015; (**c**) livestock products during 1995–2015.

**Figure 6 ijerph-19-15774-f006:**
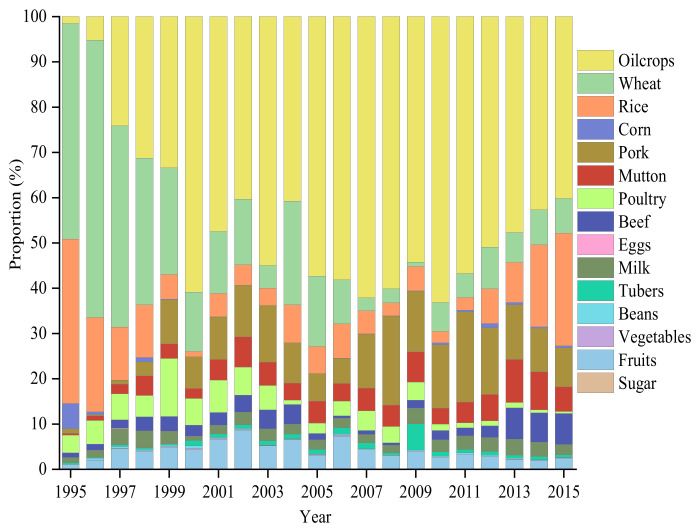
Compositions of GHG emissions embodied in imports by the type of agricultural products during 1995–2015.

**Figure 7 ijerph-19-15774-f007:**
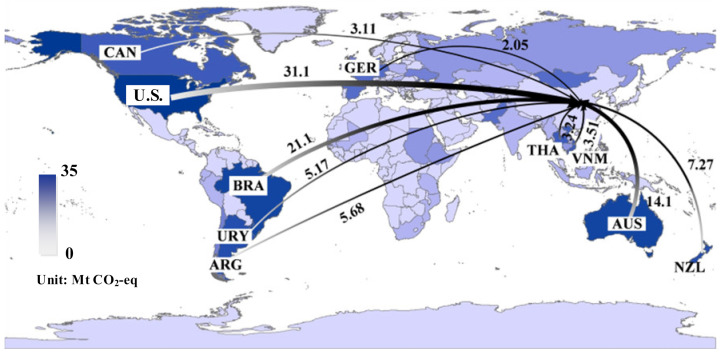
Major GHG emission flows embodied in China’s import of agricultural products in 2015.

**Figure 8 ijerph-19-15774-f008:**
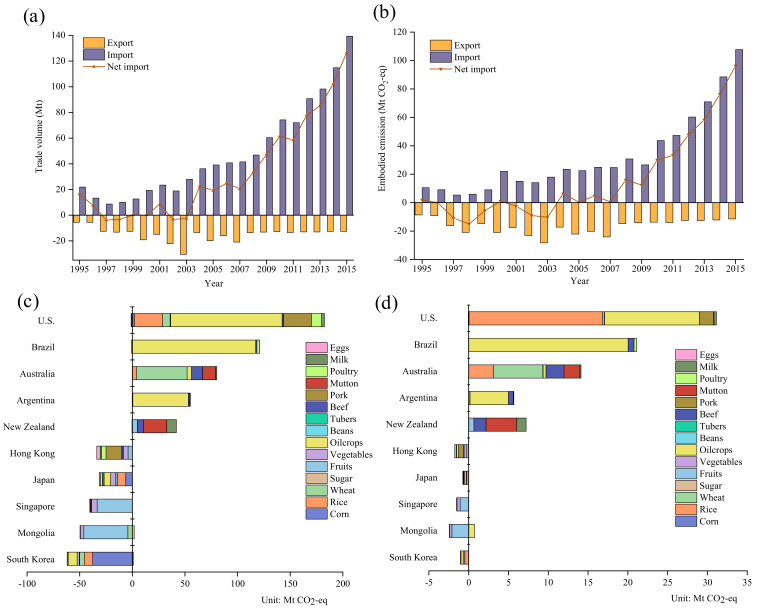
(**a**) China’s trade of agricultural products during 1995–2015; (**b**) GHG emissions embodied in China’s trade of agricultural products during 1995–2015; (**c**) major trading partners in terms of China’s *EEB* during 1995–2015; (**d**) major trading partners in terms of China’s *EEB* in 2015.

**Figure 9 ijerph-19-15774-f009:**
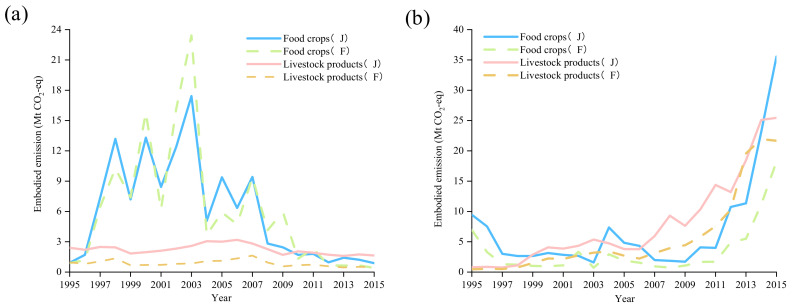
(**a**) GHG emissions embodied in China’s export of main agricultural products under different CEFs; (**b**) GHG emissions embodied in China’s import of main agricultural products under different CEFs. Note: F represents the results based on the CEFs of FAO and J represents the result based on the CEFs of Hawkins et al. [38].

**Table 1 ijerph-19-15774-t001:** China’s supply and demand of major agricultural products in 2015.

Item	Domestic Production (Mt)	Domestic Consumption (Mt)	Supply Rate
			Production/Consumption
Grain	563.1	469.3	112.0%
Fruit	271.0	265.9	101.9%
Vegetables	769.2	840.3	91.5%
Meat	84.5	86.1	98.2%
Milk	38.7	43.6	88.9%
Sugar	11.6	15.6	74.4%
Oil crops	24.0	68.7	35.0%
Beans	11.8	87.7	13.4%

## Data Availability

The data presented in this study are available in the Appendix A.

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
