# Peer review of "China’s Trade of Agricultural Products Drives Substantial Greenhouse Gas Emissions"

_ijerph, 2022, doi:10.3390/ijerph192315774_

Round 1

Reviewer 1 Report

First, I would like to mention that, in my opinion, the theme, current content and analysis of the manuscript are indirectly and partially related to the scope of the journal in the field of environment aspect and public health. In my opinion, the manuscript of “China’s Agricultural Product Trade Drives Substantial Green-House Gas Emissions” has done a lot of work, and the research conclusion is of practical significance. 

Second, in my view, the topic is interesting, and the authors’ research is valuable; however, it contains several shortcomings and inaccuracies that should be eliminated and solved before further processing the manuscript.

(1) In 2020, China has put forward the goal of dual carbon, meanwhile,international trade frictions have been intensifying in recent years. The author also mentioned in the manuscript. But, the data used in this paper comes from the 1995-2015 China Customs Statistics Yearbook, this data is relatively old, whether the research conclusions can accurately represent the current situation, whether the data can be updated.

(2) The manuscript reflects the author's sufficient workload, has done a lot of measurement work, but the conclusion of the paper is slightly insufficient. I suggest that the author should make more discussion on the application value of the achievements and   supplementing the trend of agricultural trade between different countries and adjusting trade policies in a timely manner.

Reviewer 2 Report

Minor revision, please see the attachment.

Reviewer 3 Report

This paper is well reading and contain some meaningful results, some suggestions to improve the quality.

1. Why use data from 1995-2015, as it's 2022, it maybe no meaning for nowadays

2.Figure 1 and 2 contain too many information and I suggest divide these figures

3.Which GHG emission  is caculated? CO2, CH4, N2O, or other emission? the author should give out in this paper

Round 2

Reviewer 3 Report

The author have solved all the questions and problems